# Limiting motorboat noise on coral reefs boosts fish reproductive success

Sophie L. Nedelec [1✉], Andrew N. Radford [2], Peter Gatenby[3], Isla Keesje Davidson[2], Laura Velasquez Jimenez[3], Maggie Travis[4], Katherine E. Chapman [1], Kieran P. McCloskey[1], Timothy A. C. Lamont[1,5,6], Björn Illing[3,7], Mark I. McCormick[3,8] & Stephen D. Simpson[1,2]

Anthropogenic noise impacts are pervasive across taxa, ecosystems and the world. Here, we experimentally test the hypothesis that protecting vulnerable habitats from noise pollution can improve animal reproductive success. Using a season-long field manipulation with an established model system on the Great Barrier Reef, we demonstrate that limiting motorboat activity on reefs leads to the survival of more fish offspring compared to reefs experiencing busy motorboat traffic. A complementary laboratory experiment isolated the importance of noise and, in combination with the field study, showed that the enhanced reproductive success on protected reefs is likely due to improvements in parental care and offspring length. Our results suggest noise mitigation could have benefits that carry through to the population-level by increasing adult reproductive output and offspring growth, thus helping to protect coral reefs from human impacts and presenting a valuable opportunity for enhancing ecosystem resilience.

[1] University of Exeter, Biosciences, Hatherly Laboratories, Prince of Wales Road, Exeter EX4 4PS, UK. [2] University of Bristol, School of Biological Sciences, Life Sciences Building, 24 Tyndall Avenue, Bristol BS8 1TQ, UK. [3] ARC Centre of Excellence for Coral Reef Studies, James Cook University, Townsville, QLD 4811, Australia. [4] Biology Department, University of Puget Sound, 1500N. Warner Street, CMB 1088 Tacoma, WA, USA. [5] Australian Institute of Marine Science, Perth, WA 6009, Australia. [6] Lancaster Environment Centre, Lancaster University, Library Avenue, Lancaster LA1 4YQ, UK. [7] Thünen Institute of Fisheries Ecology, Herwigstr, 31, 27572 Bremerhaven, Germany. [8] Coastal Marine Field Station, School of Science, University of Waikato, Tauranga, New Zealand. ✉email: s.nedelec@exeter.ac.uk

Anthropogenic noise is a major global issue listed among the top environmental risks to human health[1], and a serious concern for wildlife including mammals, birds, amphibians, fishes and invertebrates[2]. Noise pollution can cause stress, distraction, masking and injury, leading to disruption at all levels of biological organisation[2,3]. The rising tide of noise that threatens humans and wildlife means rapid solutions with wide-reaching impact are needed[1,2]. Transportation is the primary source of noise pollution above and below the water[1,4], and noise emissions from traffic have widespread detrimental impacts across taxa and ecosystems[2]. However, traffic noise pollution can be rapidly reduced by shifts in human behaviour, as was observed during recent decreases in activity associated with the COVID-19 pandemic[5]. Reducing noise pollution has potential benefits for wildlife, especially in ecologically vulnerable areas.

The biodiversity and socioeconomic value of coral reefs, combined with their vulnerability, make them a high priority for resilience-based management[6]. Coral reefs are highly biodiverse but are among the most threatened ecosystems in the world; half of coral reefs have already become dominated by algae (a 'catastrophic change in state'[7]). At least 500 million people worldwide maintain socio-ecological relationships with, and depend on, coral reefs for a suite of ecosystem goods and services, often requiring access via motorised boats[8,9]. Coral reef ecosystem health is reliant on functionally diverse fish communities[10] but on coral reefs, motorised traffic is a growing source of noise pollution that threatens fishes throughout their life cycle. Noise impacts development, orientation and interactions between conspecifics and heterospecifics[11–14]. Some studies have even shown direct adverse consequences of traffic noise for survival and reproductive success[13,14]. However, there is hope: avoiding driving near coral reefs reduces noise exposure and could enhance fish fitness, although experimental tests are lacking. The resilience of coral reef systems depends on the ability of fish populations to recover from shocks such as heatwaves and hurricanes, making reproductive success a critical measure of effective management. Here, using an established model system on the Great Barrier Reef, we test the hypothesis that traffic management over an entire breeding season (3 months) could result in improved fish reproductive success, parental-care behaviour and juvenile size.

## Results

We monitored breeding, brood survival and offspring size of wild spiny chromis (*Acanthochromis polyacanthus*), a common planktivorous damselfish that exhibits parental care of offspring until the end of the juvenile phase (details in Supplementary Information). We created two types of experimental site: 'limited-boating' reefs (where we requested motorboat drivers avoid reefs, or approach slowly and anchor further than 20 m when accessing them) and matched-control 'busy-boating' reefs (where we drove motorboats for ~1.25 h per day); details in Supplementary Information. Similar numbers of spiny chromis pairs produced offspring on limited-boating ($N = 46$) and busy-boating ($N = 40$) reefs. However, limited-boating pairs were almost twice as likely to have surviving offspring at the end of the breeding season (proportion of nests, Chi-square test: $X^2_1 = 4.67$, $p = 0.031$; Fig. 1A).

To explore potential reasons for this greater end-of-season success on limited-boating reefs, we combined data from our field experiment with those from a complementary laboratory experiment. The latter compared responses to busy-boating playback and no-boating playback. Our laboratory experiment allowed isolation of noise as the causal factor of impacts on reproductive success and, since eggs are naturally laid in caves within the reef, assessment of effects on clutch characteristics, egg development and parental-care investment that were unobservable in the wild. Limited-boating

pairs were not more successful because of a difference in timing of breeding (wild, linear mixed-effects model (LMM): $X^2_1 = 1.23$, $p = 0.268$; Table S1A; captivity, two-sample Welch's $t$-test: $t_{19} = 0.55$, $p = 0.591$), in clutch size (captivity: $t_{10} = 0.11$, $p = 0.917$) or in the number of predators around nests (wild, generalised linear mixed-effects model (GLMM): $X^2_1 = 1.39$, $p = 0.239$; Table S1B). Nor was the greater nest success the consequence of a difference in hatching success: in captivity, hatching success was equivalent between treatments (two-sample Welch's $t$-test: $t_{13} = 0.71$, $p = 0.493$); and, in the wild, the trend was for fewer larvae per brood on limited-boating reefs (mean $\pm$ SE $= 113 \pm 12$) compared with busy-boating reefs ($139 \pm 9$) the first time they were counted ($t_{53} = 1.81$, $p = 0.076$). Instead, the greater end-of-season reproductive success on limited-boating reefs was the consequence of improved post-hatching survival. In the wild, offspring survival (determined every 4 days) was affected by the interaction between treatment and hatch count: on limited-boating reefs, survival was better for smaller broods than larger broods, whilst on busy-boating reefs smaller broods had lower survival than larger broods (Cox model: $X^2_1 = 40.79$, $p < 0.001$). There was better offspring survival overall on limited-boating compared with busy-boating reefs ($X^2_1 = 34.87$, $p < 0.001$; Table S1C; Fig. 1B). The smallest wild broods that hatched in the busy-boating treatment may have suffered complete mortality before they were first observed due to their greater vulnerability to noise[13]. This could explain the bias towards higher number of hatchlings counted at the first observation in the busy-boating treatment in the wild, not observed in the laboratory where counts of hatchings always occurred within hours of hatching. In captivity, there was a congruent trend towards better survival (determined at 21 days post-hatching) in the no-boating treatment compared with the busy-boating treatment (Wilcoxon ranked-sums test: $W = 87$, $N_{\text{busy-boating}} = 9$, $N_{\text{no-boating}} = 13$, $p = 0.060$). Whilst predation is the strongest direct driver of juvenile coral reef fish survival in the wild, with cannibalism occurring rarely in study species[15,16], the laboratory trend suggests other or additional drivers. Better survival in the limited-boating treatment compared with the busy-boating treatment could be due to reduced stress and consequent impacts on development[11,17].

Limited-boating reefs not only had more surviving offspring than busy-boating reefs, but those offspring were larger. We measured the standard length of up to ten individuals per brood approximately weekly and found that limited-boating juveniles were longer than those from the busy-boating treatment (LMM, treatment*age interaction: $X^2_1 = 8.97$, $p = 0.033$; Table S1D; Fig. 1C). The same interaction effect of treatment (no-boating vs busy-boating) and age was found in captivity, where we measured up to 10 individuals per brood (kept in isolation from their parents to avoid competition for, or provisioning of, food) at days 21 and 42 post-hatching ($X^2_1 = 7.95$, $p = 0.005$; Table S1E; Fig. 1D). Although hatchling dry weight increased from day 21 to day 42 in captivity, there was no significant effect of either treatment ($X^2_1 = 0.12$, $p = 0.730$) or its interaction with age ($X^2_1 = 1.58$, $p = 0.209$; Table S1F). Weight as a measure of development may be subject to high variability due to food in the gut. Longer offspring are likely to survive better as there is strong size-selective mortality of larvae resulting from gape-limited predators[18], and work on coral reef fish larvae shows that species with slower pre-settlement growth are selectively preyed upon[19]. The same effect was seen whether juveniles were cared for by parents (in the wild) or not (in captivity), indicating direct effects of noise mitigation on hatchlings that may also be seen in coral reef fish species without parental care through the juvenile phase[20].

To consider direct effects at the embryonic stage, we sampled ten eggs per captive clutch within 2 h of laying. At laying, egg area, yolk sac area and dry weight (Table S1G–I), as well as embryonic developmental time (two-sample Welch's $t$-test: $t_{14} = 0.62$, $p = 0.273$), were all similar between sound treatments.

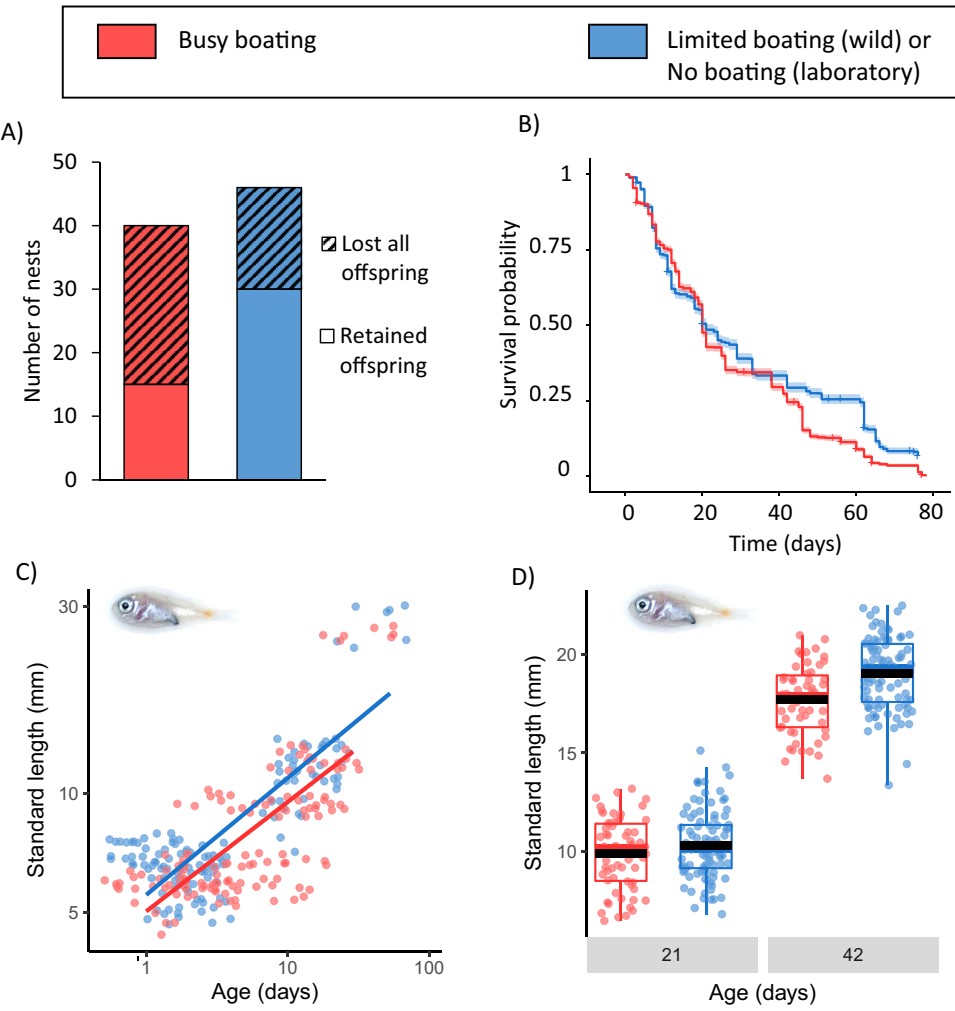

**Fig. 1 Difference in juvenile survival and length of spiny chromis exposed to busy boating or protected from motorboat noise. A** Likelihood of wild nests that successfully hatched young retaining juveniles at the end of the season (Chi-square test: $X^2_1 = 4.67$, $p = 0.031$). **B** Survival of juveniles in the wild from hatching to end of the season (Kaplan-Meier survival curve based on GLM model predictions; Cox survival model used for statistical testing: $X^2_1 = 34.87$, $p < 0.001$). Busy-boating nests = 32, limited-boating nests = 27). Standard length of 10 juveniles per brood **C** in the wild (LMM, treatment*age interaction: $X^2_1 = 8.97$, $p = 0.033$, effect size mean ± SE = 0.004 ± 0.001 mm/day, busy-boating nests = 11, limited-boating nests = 11, lines fitted for illustrative purposes using geom_smooth(method = 'lm') in R) and **D** in captivity (LMM, treatment*age interaction: $X^2_1 = 7.95$, $p = 0.005$; effect size of interaction between age and treatment mean ± SE = 0.09 ± 0.04 mm/day, busy-boating broods = 9, no-boating broods = 13, boxes show median and interquartile range (IQR), whiskers extend 1.5*IQR above or below the box, black bars show means).

However, egg area increased more during development in the no-boating treatment than in the busy-boating treatment (LMM, treatment*age interaction: $X^2_1 = 33.69$, $p < 0.001$; Table S1J). This suggests that noise affected the growth of embryos within eggs, either directly or indirectly via parental-care behaviour. Using 10 embryos sampled from each clutch 10 days post-fertilisation (i.e., the end of the embryonic phase), we also found that those from the no-boating treatment were longer (dorsal spine length: $X^2_1 = 8.10$, $p = 0.004$; Table S1K; Fig. 2A), with lower resource use (yolk sac area: $X^2_1 = 11.19$, $p < 0.001$; Table S1L; Fig. 2B). Limiting motorboat noise in our experiment had a positive effect on embryonic development that is consistent with previous studies that identified stress, disrupted tissue formation, tissue damage and altered gene expression as potential mechanisms of noise impact[11,17,21]. This may explain the differential survival that we observed; for instance, larger yolk sacs have been shown to correlate with improved survival[21]. Size-dependent embryonic mortality or selective removal of eggs by parents could explain the different lengths and survival of offspring later in juvenile development.

Both spiny chromis parents oxygenate (by fanning) and guard (by inspecting and chasing threats) their eggs;[22] thus, parental care influences offspring survival and growth[23,24], which could be improved by a reduction in noise. On day 10 of embryonic development, we examined egg-fanning and activity (as a proxy for guarding) of captive parents before and during exposure to either motorboat-noise playback (in the busy-boating treatment) or a different ambient-reef playback (in the no-boating treatment). Comparing quiet periods in the two sound treatments, parents spent a similar amount of time egg-fanning (two-sample Welch's $t$-test: $t_{13.2} = 0.01$, $p = 0.994$; Fig. 2C), and showed a trend towards greater activity in the no-boating treatment ($t_{20.0} = 1.91$, $p = 0.071$; Fig. 2D). However, at the onset of the short-term playback exposure, there was a decrease in egg-fanning ($t_{18.0} = 4.24$, $p < 0.001$; Fig. 2C) and an increase in activity ($t_{16.7} = 2.51$, $p = 0.023$; Fig. 2D) in motorboat-noise playback compared with ambient-reef playback. The short-term changes in parental behaviour during motorboat-noise exposure were present despite several weeks where individuals could have habituated or developed greater tolerance to the intermittent noise exposure. However, increased tolerance to noise overtime was

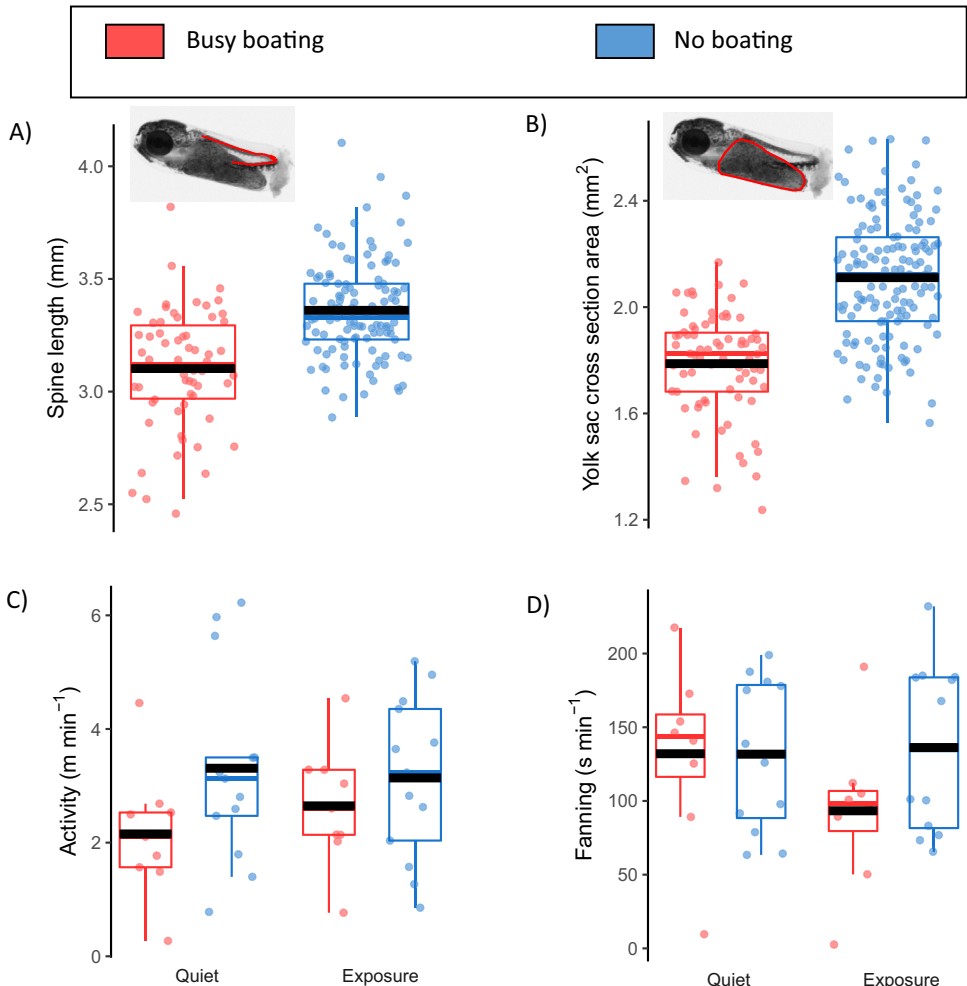

**Fig. 2 Difference in embryonic development and parental-care behaviour in captive spiny chromis exposed to intermittent motorboat noise throughout development (busy-boating treatment in red) or protected from exposure to motorboat noise (no-boating treatment in blue).** Embryonic development: **A** spine lengths were 0.23 ± 0.08 mm (7.4%) longer (LMM: $X^2_1 = 8.10$, $p = 0.004$) and **B** yolk sac areas were 0.30 ± 0.08 mm$^2$ (17%) larger (LMM: $X^2_1 = 11.19$, $p < 0.001$) at the end of the embryonic phase in no-boating compared with busy-boating. Parental care in quiet conditions for each rearing condition compared with exposure to motorboat playback if reared in busy-boating (red) or a different ambient track if reared in no-boating (blue): **C** no difference in fanning in quiet periods (two-sample Welch's t-test: $t_{13.2} = 0.01$, $p = 0.994$) but the onset of motorboat noise decreased fanning (mean difference of 9 s min$^{-1}$, two-sample Welch's t-test: $t_{18.0} = 4.24$, $p < 0.001$). There was a **D** trend towards greater activity in no-boating compared with busy-boating in quiet periods (mean difference = 1.16 m min$^{-1}$, two-sample Welch's t-test: $t_{20.0} = 1.91$, $p = 0.071$), then activity in busy-boating increased at the onset of motorboat noise to reach a level similar to that in no-boating (mean difference = 0.67 m min$^{-1}$, two-sample Welch's t-test: $t_{16.7} = 2.51$, $p = 0.023$). In all panels boxes show median and interquartile range (IQR), whiskers extend 1.5 * IQR above or below the box, black bars show means. Busy-boating broods = 9, no-boating broods = 13.

not observed, in line with previous work[13]. Egg-fanning results in increased embryonic oxygen consumption, faster development and general promotion of brood success[23]. Therefore, the longer embryos in the no-boating treatment could be explained by protection from noise-induced interruptions in parental egg-fanning. The energetic costs of elevated activity during noise exposure may require compensatory rest, which could explain our observed reduction in activity during quiet periods in the busy-boating treatment. We did not observe compensatory egg-fanning during quiet periods. Therefore, the increased size and survival of end-of-season offspring on limited-boating reefs could be partially explained by reduced disturbance to parents during the embryonic phase.

## Discussion

Using complementary studies that combine ecological validity in the wild with tight experimental control in the laboratory, we show that protecting coral reefs from motorboat activity can boost the reproductive success of adults and the length of juveniles in fish. By limiting motorboat activity around reefs, the number of spiny chromis nests producing viable offspring was almost doubled, offspring grew faster and their survival within nests was improved. In our laboratory study isolating sound as the disturbance, protection from motorboat noise led to increased parental egg-fanning, lower use of energy reserves during embryonic development, and improved offspring survival. There is no reason to suspect that limiting traffic noise would only benefit our study species: chronic and acute anthropogenic noise compromises behaviour, physiology, reproduction and survival in a range of marine and terrestrial organisms, while natural, unpolluted soundscapes are key to settlement, recruitment and other ecological functions[3]. It is true that some fishes show increased tolerance to noise disturbance when faced with repeated exposure (e.g. [25–27]), but this is not seen in all, including our study species (e.g. [13,28]). There is mixed evidence from terrestrial systems on ecological recovery in quieter conditions (e.g. birdsong in lockdown increased[28,29], while seed dispersal around disused gas wells

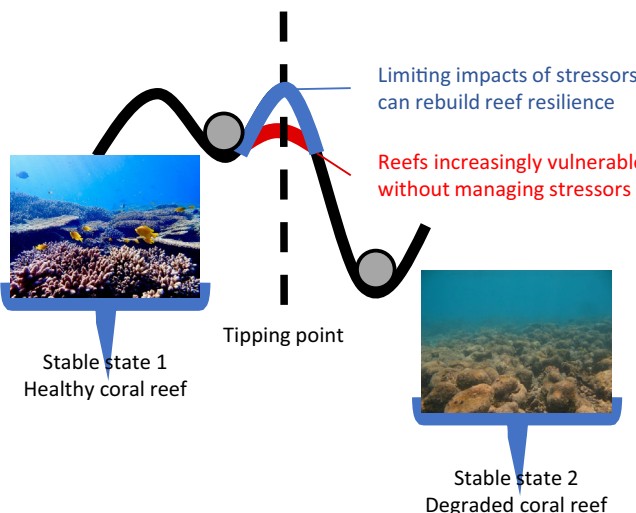

**Fig. 3 The 'ball and cup' analogy of resilience, where greater resilience equates to a deeper cup, shows that building resilience (up the dashed line) in a system makes it less likely to be pushed past a tipping point into an irreversible change in state, deemed a 'regime shift'.** Perturbations that may push the system towards its tipping point include hurricanes, heatwaves, disease outbreaks and over-exploitation. Resilience-based management focuses on minimising local anthropogenic stressors[7]. In the context of coral reefs, motorboat noise is a local anthropogenic stressor that can be managed. Photo credit: S Simpson.

did not[30]) but, in general, reproductive success and survival are strongly linked to population stability[31]. In fish in particular, improvements in energy intake and energy expenditure could be important drivers of population growth via size-dependent fecundity[32]. Therefore, limiting traffic noise could help conserve and restore healthy populations of coral reef fishes.

Three pillars of action are required to rebuild coral reefs worldwide: (1) reducing climate threats, (2) reducing local threats and (3) active restoration[33]. Globally, one million species are at risk of extinction[34] and human-induced regime shifts (Fig. 3) are increasingly tipping ecosystems towards states that are lower in biodiversity, with increased likelihood when resilience is low[35]. Whilst actions such as the emission of waste and pollutants can lead to loss of resilience[36], limiting local and predictable threats and conserving healthy populations can build back resilience, increasing the likelihood of recovery from shocks such as heatwaves and hurricanes and lessening the likelihood of state-change[7,35]. The scale of societal change required to achieve the goals of the IPCC and the UN Sustainable Development Goals fully is immense and will take longer than the intervals between anthropogenically induced shocks[37,38]. Rapid, evidence-based local interventions that increase resilience can 'buy time' by improving recovery potential from stochastic events[31,35]. Limiting traffic noise on coral reefs provides an opportunity to build resilience in these threatened ecosystems, and represents an example of the adaptive management increasingly recognised as a critical way of maintaining valuable social-ecological systems[31,35].

## Methods
**Permits and ethics approval.** Animal collections and all experimental procedures were conducted with ethical approval from the University of Exeter (2013/247), James Cook University (A2361) and Lizard Island Research Station, permission from the Great Barrier Reef Marine Park Authority (GBRMPA) (G17-39752.1) and under licence from the Australian Government Department of Fisheries (170251).

**Study species.** The spiny chromis (*Acanthochromis polyacanthus*) is a damselfish that exhibits bi-parental care of eggs and juveniles at nests within shallow reef

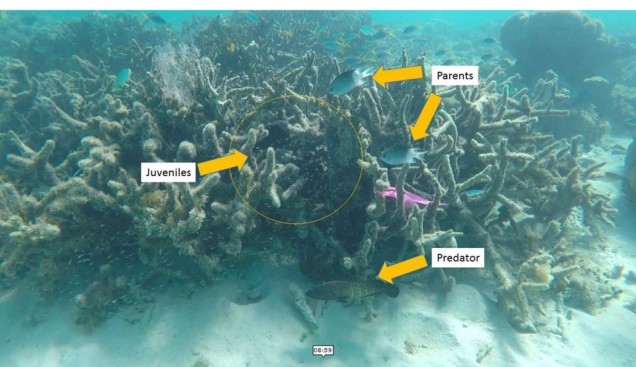

**Fig. 4 Spiny chromis (*Acanthochromis polyacanthus*) nest at the edge of coral reef habitat.** Parents, juveniles and a predator (peacock grouper, *Cephalopholis argus*) can be seen in this photo. Photo credit: S Nedelec.

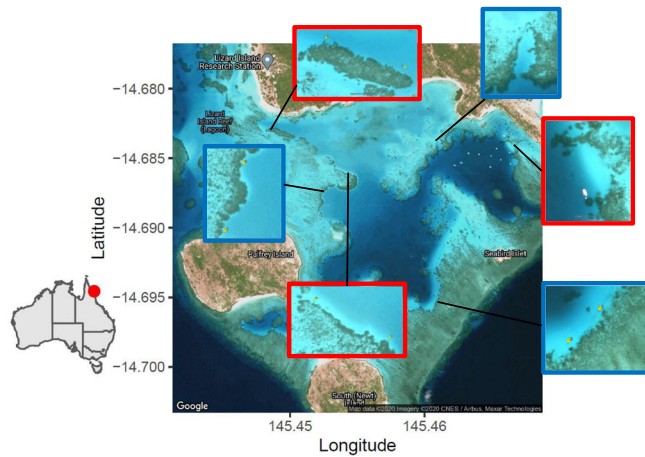

**Fig. 5 Map of the experimental sites.** Red sites were 'busy-boating' areas while blue sites were 'limited-boating' areas.

habitat in the tropical Western Pacific[39] (Fig. 4). Spiny chromis are planktivores on the Great Barrier Reef, importing nutrients from the plankton to the reef[40]. As such, their preferred habitat is the reef edge (within ~7 m). Most pairs raise one clutch per season—in our study, second clutches occurred in only 7% of the population—so we tracked first clutches from adult pairs. Adults enhance their reproductive success by fanning eggs to oxygenate them, and chasing away potential predators from eggs and juveniles[16,41]. Adults also allow their offspring to eat some of their body mucus in a behaviour known as 'glancing' that has potential nutritional and/or immunological benefits[42].

*Field study.* We conducted the field study at Lizard Island Research Station (LIRS) (14° 40′ S, 145° 28′ E), Great Barrier Reef, Australia over an entire spiny chromis breeding season (90 days from 23 October 2017 to 20 January 2018).

**Sites and nests.** We selected six coral reef edge sites (113–218 m in length) within the lagoon on the south of Lizard Island (Fig. 5). Water temperature ranged from ~26 °C at the start of the season to ~29 °C at the end of the season. The reefs were composed mainly of a mixture of live and dead coral. Prevailing currents were wind-driven from south to north. Three of the sites were exposed to 'busy boating' while boating was limited at the other three. Treatments were allocated to sites partly randomly and partly allowing for ease and safety of motorboat access. Nest positions within sites did not differ between treatments in: depth of bottom next to reef (1–5 m at mid-tide with a tidal range of ~2 m, $N = 67$ measurements at a range of tidal heights, mean ± SE depth = 2.3 ± 0.1 m; LMM, sound treatment: $X^2_1 = 0.82$, $p = 0.365$, random effect of site: variance = 0.14, standard deviation=0.38); nest height above the sand (range = 0–4 m, mean ± SE = 0.6 ± 0.1 m; sound treatment: $X^2_1 = 0.52$, $p = 0.470$, site: variance=0.33, standard deviation = 0.57); or distance from the edge of the reef (range = 0–6.6 m, mean ± SE = 1.2 ± 0.2 m; t-test: $t_{33,32} = 1.61$, $p = 0.112$; a linear model did not fit the data). A small number of broods (mean ± SE per site = 4.7 ± 1.0) were already present at each site at the start of the season; spiny chromis occasionally breed outside the main breeding season. These were marked and ignored for the purpose of the experiment. The mean ± SD number of nesting pairs identified at each site at the start of the season was 9.5 ± 2.0 for limited-boating sites and 10.7 ± 3.2 for busy-boating sites. Nesting pairs

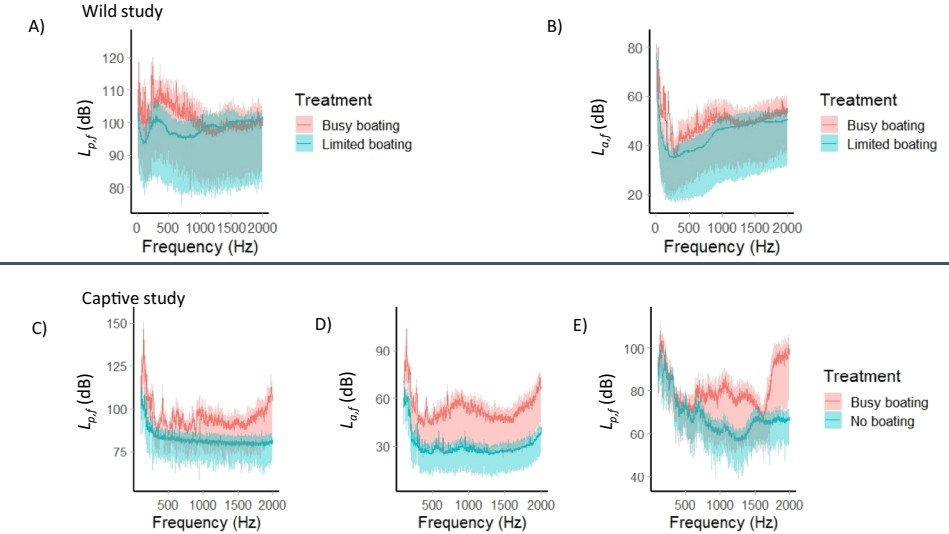

**Fig. 6 Pressure power spectral density level ($L_{p,f}$ (re 1 µPa$^2$ Hz$^{-1}$)) and particle acceleration power spectral density level ($L_{a,f}$ (re 1 (µm s$^{-2}$)$^2$ Hz$^{-1}$)) plots showing the mean (solid line) with 5% and 95% exceedance levels (coloured band around solid line) for busy-boating and limited-boating or no-boating treatments.** **A** $L_{p,f}$ in the wild study (average from three nests per site, 'busy boating' includes three passes of a motorboat at 10–250 m per recording location, recordings of limited boating were three minutes in duration per nest). **B** $L_{a,f}$ in the wild study (same recording design as for pressure). **C** $L_{p,f}$ in the parental tanks in the captive study (27 locations in a 3 × 3 × 3 grid within the tank, 1-min sample of motorboat playback and ambient reef sound playback for each case). **D** $L_{a,f}$ in the parental tanks in the captive study (same recording design as for pressure), **E** $L_{p,f}$ in the juvenile tanks in the captive study from the centre of the rearing tank (these tanks were too small to accommodate the particle motion sensor).

did not always form clear, stable pairs with obvious territories and so pairs without broods were not tracked through the experiment. The mean ± SD total number of adults at each site at the start of the experiment was 185 ± 92 for limited-boating sites and 119 ± 20 for busy-boating sites.

**Motorboat traffic exposure and protection.** There is a navigable channel through the lagoon where the experiment was conducted. Fishing boats, tourist boats and research station boats pass through the channel, but the main source of traffic is research station boats. We chose six sites along the navigable route and randomly allocated these to treatments. Following random allocation, two sites were switched for safety reasons for motorboat drivers (Fig. 5). We experimentally elevated motorboat noise at three of the six sites (busy-boating treatment) to mimic typical traffic around a port, harbour or regularly visited reef. At these sites, we drove eight different 5 m aluminium motorboats with 40 hp Suzuki four-stroke outboard engines repeatedly along the length of the site within 10–30 m of the edge of the reef. Busy-boating sites received an average of 180 motorboat passes each day during 3–6 'exposure periods' lasting 15–20 min each; this totalled 1.25–1.5 h per day of traffic noise at each busy-boating site. The other three sites were protected from motorboat traffic (limited-boating treatment), by marking these reefs on the research station map as areas to avoid by at least 100 m and monitoring activity in the lagoon daily. When experimenters needed to access protected sites, speed was reduced to that where no wake was created (roughly ¼ throttle) within 100 m and boats were anchored 20 m from the reef. See Supplementary Information for further details of motorboat traffic exposure and protection.

**Acoustic recordings and analysis.** We made acoustic recordings of both pressure and particle motion at three locations within each site using an accelerometer with integrated hydrophone (M20-040 manufactured and calibrated by Geospectrum Techologies Inc. Dartmouth, Canada; sensitivity follows a curve from 0 to 5000 Hz) and a digital recorder (Zoom F4, Zoom Corporation, Tokyo, Japan; calibrated using pure sine waves measured with an oscilloscope). See Fig. 6 and Supplementary Information for further details of acoustic recordings, analysis and results.

**Breeding and reproductive success.** Each site was checked by snorkellers every other day to monitor breeding by spiny chromis pairs. Nests with new hatchlings were marked with flagging tape and continued to be checked every other day throughout the season to monitor reproductive success. The day in the season that broods hatched was used to test for an effect of motorboat exposure on the timing of breeding using a linear mixed-effects model (fitted in R) with site as a random effect. The proportion of nests retaining offspring at the end of the season in the two treatments was compared using a Chi-squared test.

**Brood size.** We counted the number of offspring in broods within four days after hatching at a subset of 59 nests; 32 in the three busy-boating sites ($N = 9$, 11, 12) and 27 in the three limited-boating sites ($N = 5$, 10, 12). Average clutch size at

hatching in the wild was 126 ± 16 (mean ± SE). Some of these nests were part of the predator presence observations (details below), some were part of the size monitoring (details below) and some were independent. We counted the number of offspring in three photos and used the highest number for analyses. We tested for an effect of motorboat treatment on brood size at hatching using a Welch's $t$-test.

**Predator presence around nests.** We determined baseline predatory threat (counts of heterospecific piscivores, potential predators of juveniles) using video camera (GoPro 5) deployments. Thirty-three nests (not studied for size) were videoed once or twice between 1 and 11 days post-hatching; 18 in the three busy-boating sites ($N = 7$, 6, 5) and 15 in the three limited-boating sites ($N = 6$, 5, 4). A total of 55 videos were analysed. A camera stand was placed at each nest on the first or second day post-hatching and remained in place 2–3 m from the nest. For each survey, after GoPro cameras were attached to camera stands, several minutes settling time was allowed (mean ± sd: 827 ± 35 s), followed by a 30-s recording of predator presence in the absence of any motorboats. All nests that were videoed were at least 10 m away from one another (parents spend most of their time within 2 m of the nest). Videos were randomly named and analysed by KEC without sound (to remain 'blind' to treatment) using *BORIS 7.6.1*[43]. Spiny chromis offspring are small and vulnerable to any piscivore on the reef and parents defend their offspring by chasing potential predators. The number of heterospecific piscivores at each nest was surveyed (conspecifics are known to cannibalise offspring, but this is very rare[16]). A negative binomial regression parameterised such that the variance is a quadratic function of the mean was fitted using *glmmTMB* in R with piscivore counts as the response variable, motorboat treatment and days into the season as potentially interacting fixed factors, and nest and site as random effects.

**Juvenile survival.** It was not possible to observe the eggs as they were laid in caves, so we studied juveniles from when they could be observed above the substrate (shortly after hatching – this species completes the larval stage inside the egg and hatches at the juvenile stage[16]). We also counted the number of surviving offspring at the subset of 59 nests every 4–8 days. When all juveniles from a nest could be captured in a frame, we used three photos per time point and used the highest number. As juveniles aged, they used more space and could not be captured in a single photo; then, they were counted by snorkellers with experience in fish surveys (SLN and IKD). The reliability of snorkellers' counts was tested against one another and did not differ when there were <20 juveniles and were accurate to the nearest 5 when there were >20 juveniles based on counts of 43 nests. Usually, however, when there were >20 juveniles at a nest, they were at an earlier developmental stage and counts could be taken from photos. Survival of juveniles was recorded as number of days from hatching until they were no longer seen at the nest. Where all offspring from a nest were apparently lost to predation (mean survival time = 21 days), the nest continued to be monitored every other day for the remainder of the season to ensure offspring had not temporarily disappeared and to check for second clutches. A Cox proportional-hazards survival model was fitted in R with motorboat treatment and initial hatching count as fixed effects, and nest and site as random

effects. We discounted three nests where counts increased due to assumed experimenter error or immigration. The package *Coxme* was used in *R* to test for the interaction between treatment and start count (hatch day was not included in the model as there was no indication of an effect of treatment or site on hatch day), with nest and site as random effects. The package *coxph* was used to create Fig. 1B, which does not account for the random effects, but is used for illustrative purposes.

**Juvenile size**. We monitored juvenile size at a subset of 22 nests; 11 in the three busy-boating sites ($N = 3, 4, 4$) and 11 in the three limited-boating sites ($N = 3, 4, 4$). Up to 10 juveniles (depending on catch success) were caught by snorkellers or SCUBA divers using hand nets from each nest each week. A total of 275 juveniles between 1 and 53 days post-hatching were sampled. Juveniles were transported to the field station in bags of fresh seawater. We measured standard length either under the microscope at 10× magnification or with a Vernier caliper, depending on fish size. All nests within a site were sampled on the same day and each site was visited for juvenile size sampling each week. Data were log-transformed and analysed using a linear mixed-effects model (fitted in *R*), with age and motorboat treatment as fixed effects, and nest and site as random effects.

*Laboratory study*. We conducted the laboratory study in the Marine and Aquaculture Research Facilities Unit (MARFU) at James Cook University, Townsville, Australia from March to July 2018. Spiny chromis adults were caught with fine monofilament barrier nets and hand nets from the section of reef on the lagoon side of Palfrey Island within the lagoon around Lizard Island in the northern Great Barrier Reef (14° 41′ S, 145° 27′ E) during November 2016. All adults would have experienced equivalent prior noise exposure. The mean ± SE standard length of adults was $10.7 ± 0.1$ cm. Spiny chromis were randomly allocated to treatments and were housed in 30 male–female pairs, and maintained at a mean ± SE temperature of $27.7 ± 0.1$ °C in the presence of either a busy-boating treatment (playback of four of the five recorded motorboats in a pattern matching exposures in the field) or a no-boating treatment (playback of ambient reef sound). We kept most of each brood with the parents to measure survival (cannibalism can rarely occur in this species under stress[16]) and isolated 50 individuals per brood as a single group in a separate tank (where parents could not compete with offspring for food) with the same playback treatment to measure size. See Supplementary Information for further details of tank setup and conditions, acoustic exposure regime, playback construction, acoustic recording analysis and results for the tanks.

**Breeding**. We checked all adult tanks daily after lights were switched on, but before playbacks began, for the presence of a newly laid clutch. The number of days since the start of the treatment that clutches were laid was used to test for an effect of motorboat noise exposure on the timing of breeding using a two-sample Welch's *t*-test.

**Clutch size and brood size**. We photographed newly laid clutches and estimated clutch size by counting the number of eggs in a square on an overlaid grid and counting the number of grid squares containing eggs. All adult tanks were checked daily after lights were switched on, but before playbacks began, for the presence of a newly hatched brood. Clutch size and brood size were compared between treatments using two-sample Welch's *t*-tests.

**Egg characteristics and embryonic development**. We monitored clutch-level and individual-level egg and embryo characteristics at days 1 and 10 during the egg phase of the first clutch laid by each breeding pair. Measures taken were: (1) egg area, (2) yolk sac area, (3) dorsal spine length (day 10 embryos only), and (4) dry weight (at 10 days). Egg area, yolk sac area and spine length were obtained by measuring 10 randomly sampled individuals per clutch under a light microscope (Olympus SZXY). For dry weights, fish were dried in an oven for >24 h at 60 °C and weighed on a Mettler microbalance with ±0.001 mg accuracy. The number of days between laying and hatching was used as the embryonic developmental time. Linear mixed-effects models (fitted in *R*) were used, with clutch as a random effect and motorboat treatment as a fixed effect.

**Parental care of embryos**. We filmed parental activity (distance moved by both parents) and time spent fanning eggs at day 10 of the egg phase during periods of playback. Two cameras were used: a Logitech HD Webcam C615 camera positioned 45 cm above the tank looking down with the entire tank in the field of view ('top camera'), and a GoPro HERO 5 positioned inside the tank in front and looking into the nest ('side camera'). Following a 30-minute settling time, minimising the disturbance to the fish, baseline behaviour was observed for five minutes. Then, in 'busy-boating' tanks, motorboat noise was played for a further five minutes, while in 'no-boating' tanks, a different ambient track was played for five minutes.

Two key nest-caring parental behaviours were identified for analysis:

(1) Activity – the distance travelled by parents. Average distance travelled within each breeding pair was calculated from the total distance travelled by both the male and female. Activity was observed from the top camera. Distance was calibrated by measuring a known distance on the bottom of the tank, present in all videos. The distance travelled was calculated by marking the position of the fish every second using the manual tracking feature of *ImageJ* version 1.52d (https://imagej.nih.gov/ij/download.html).

(2) Fanning – the amount of time parents spent fanning the clutch of eggs. Fanning was observed from the side camera and analysed using Solomon Coder software (https://solomoncoder.com/download.php).

*T*-tests were used to test for effects of treatment on parental care behaviour.

**Juvenile survival**. We counted the number of juveniles that survived with their parents at day 21 post-hatching (maximum count from three photos for each tank). Survival was measured at day 21 because that was the mean survival time in the field; also, by day 42 post-hatching, most parents had produced a second clutch which confounded observations of survival. The fish removed at hatching were included in the final count by modelling their survival as equal to that of the rest of the brood. Survival was converted to a percentage from the number of eggs laid in the clutch. This measure of survival is conservative compared with that expected in the field because the only potential predators of the juveniles in the tanks were their parents. Percentage survival rates were non-normally distributed and so were compared between treatments using a Wilcoxon signed-ranks test.

**Juvenile size**. We measured the standard lengths (from photos using ImageJ) and dry weights of ten juveniles per clutch at day 21 post-hatching and of eight juveniles per clutch at day 42 post-hatching, following humane sacrifice; fish were dried in an oven for >24 h at 60 °C and weighed on a Mettler microbalance with 0.001 mg accuracy. We measured length and weight from juveniles that were isolated from the parents to avoid the possibility that the parents could compete with the juveniles for food. Dry weights at day 21 and 42 were compared between treatments using an LMM with clutch as a random effect.

**General statistical approaches**. For measures of parental care, or at the level of clutch, *t*-tests or Wilcoxon signed ranks tests were used. Where we measured several individuals from within multiple sites, clutches or broods, LMMs or GLMMs were used to control for the random effects of site, clutch or brood, provided models fit the data satisfactorily. Plots of residuals vs fitted values were examined to check model fit and where models did not fit the data, standard tests such as t-test/Wilcoxon signed ranks were used in their place. Any effects among nests (such as slight variations in water flow) were therefore controlled for by the LMM or GLMM statistical models. The variance attributed to, and standard deviation of the variance for, random effects are presented as part of the full output from models. We used the same approach for model selection as in[13]. To establish the best-fitting model, terms were eliminated one by one from a maximal model. Simplified models were compared with more complex ones using maximum likelihood ratio tests that employ chi-square statistics to establish whether a simpler model performed significantly worse at explaining the data than a more complex model. If the simpler model was not significantly worse when a term was removed, the simpler model was deemed better and thus the removed term was dropped. If the simpler model was significantly worse, the term was maintained in the model[44]. The degrees of freedom from maximum likelihood tests presented in the Results of the main paper are the difference between the degrees of freedom of the simpler and the more complex models. All potential interactions of fixed effects were examined and are only presented where their exclusion from the model made the model significantly worse at explaining the data at the significance level $p < 0.10$. When an interaction was presented, the effect of the main effects was established by comparing a model with the main effect in question, although not its interaction, with a model that dropped that effect. The package 'lme4' was used for LMMs. Effect sizes and standard errors given are the results from the model summaries within the R code packages used to create models. Full model outputs are presented in Table S1.

**Reporting summary**. Further information on research design is available in the Nature Research Reporting Summary linked to this article.

## Data availability

The data generated in this study are provided as a Source Data file.

## Code availability

All code used for statistical analysis in R is standard code that is freely available online, for acoustics analysis code, see reference [45] and https://gitlab.com/RTbecard/paPAM/blob/master/README.md[46,47].

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

## Acknowledgements

Lizard Island staff, field assistants (Brendan Nedelec, Kasey Barnes, Olivia Rose, Sam Wines), MARFU staff (Ben Lawes, Simon Wever, Andrew Thompson), MARFU employees and volunteers (Ella Smyth, Geoffrey Dominic Yau, Millicent Nichols, Emily Mulroy, Hannah Wolstenholme, Blake Spady, Shannon McMahon), lab advice (Eric Fakan). This work was supported by funding from a Natural Environment Research Council Research Grant (S.D.S. and A.N.R.; NE/P001572/1), an Australian Research Council Discovery Grant (M.I.M.; DP170103372), a UKRI Strategic Priorities Fund Postdoctoral Fellowship (S.L.N.), a University of Exeter Vice-Chancellor Scholarship for Postgraduate Research (K.P.M.), a German Research Foundation (DFG) research fellowship (B.I.; IL 220/2-1) and a NERC-Australian Institute of Marine Science CASE GW4+ studentship (T.A.C.G.; NE/L002434/1).

## Author contributions

Conceived and planned the study S.L.N., S.D.S., A.N.R., M.I.M.; collected data S.L.N., P.G., I.K.D., L.V., M.T., K.P.M., B.I; supported data collection, contributed to the organisation and concepts of the study T.A.C.G., M.I.M.; data analysis S.L.N., P.G., M.T., K.E.C., B.I., with support from A.N.R., S.D.S.; S.L.N. wrote the manuscript with contributions from P.G., M.T., K.E.C., A.N.R., S.D.S.; S.L.N., K.P.M. and I.K.D. created the figures; all authors contributed to final revisions of the manuscript.

## Competing interests

The authors declare no competing interests.
