## [Peer Review File · Nature Communications]

REVIEWER COMMENTS

Reviewer #1 (Remarks to the Author):

Noise has in recent years been shown to influence the fitness of species also in aquatic ecosystems, with an increasing number of studies showing effects especially on fishes. The aim of the present study was to determine if limiting boat traffic can mitigate the negative effects that the noise is known to have. The study design is thus the opposite to that usually applied when noise is added, and the results are – unsurprisingly - in line with earlier findings. The study consequently backs up earlier results on negative effects of noise on coral reef fishes and the importance of limited/regulating noise levels. The study also attempts to determine the mechanism behind the effects of noise, through a more controlled laboratory experiment, but here the results are less clear and some overstatements are made (too strong conclusions are drawn regarding offspring survival and size). More research should be done under these controlled conditions before any clear conclusions can be drawn and compared to the results from the field.

The main limiting factors of the present study, which need to be better taken into consideration in the paper are:

- 1) No information on noise levels at the protected 'limited boating reefs' in the absence of the protection, which would ensure that the 3 protected and the 3 control sites had the same noise levels before the experimental treatments started.
- 2) Some sample sizes are quite small and the results are not always clear cut (see for example line 88). Some of the conclusions drawn are consequently too strong.
- 3) It is not possible to separate between the effects of treatment and earlier experience (in the field) regarding the variables recorded in the laboratory experiment.
- 4) Only one species was investigated, which limits the generality of the findings.

Specific comments

Line 59-62. The lack of an effect of the treatment on the number of pairs producing offspring could be because of the short duration of the experiment before this variable was recorded (compared to the duration of the experiment before offspring survival was determined).

Line 84-85. How can survival be higher on the limited-boating reefs? More explanation needs to be given. On line 76 you indicate that more offspring were hatching on busy-boating than on limited-boating reefs, and on lines 81-81 that survival increased with hatch count on busy-boating reefs, but decreased on limited-boating reefs. This would result in many more offspring surviving on busy-boating reefs, as clutch size and number of breeding pairs did not differ – have I misunderstood something?

Line 91-93. There was only a trend towards better survival in the lab experiment, and the sample sizes are very small so you cannot state that survival was higher. Be more careful in drawing your conclusions.

Line 104. If treatment influenced offspring length but not weight it appears that the shape of the juveniles were different – do you have any more information on this? If weight was not influenced by treatment, it is also not true to state that the offspring were larger, they were only longer.

Line 360- If the individual were immediately exposed to the treatments in the lab, it is likely that the eggs clutch sizes had been determined before the fish were caught and exposed to the treatments. To be able to determine effects on clutch size and egg quality, you should have bred the fish in the lab

under standardised conditions before you exposed them to the treatments. Given the present design, it is not possible to separate between the effects of the treatment and earlier experience in the field.

Reviewer #2 (Remarks to the Author):

The manuscript “Limiting motorboat noise on coral reefs boosts fish reproductive success” shows that spiny chromis fish have increased hatchling survival, as well increased hatchling size, in reefs with limited motorboat noise compared to reefs with frequent boat noise. The manuscript cleverly reframes the typical story line of how noise disturbance affects animals to one where decreased noise exposure (the old baseline) leads to improved fitness. The results of the manuscript are sound and the combination of studies in the field and in captivity enables the authors to identify the mechanisms that underlie the change in offspring survival between treatments. Having said that, I do have some comments.

My main concern is the argumentation around improved fitness with noise mitigation. As mentioned before, the manuscript is framed in such a way that the focus is on improved hatchling survival in decreased noise conditions, rather than decreased survival in increased noise conditions. I like the framing as a change in perspective for the field, but I miss some discussion on repeated or chronic noise exposure. Showing effects of noise mitigation requires that the original state was one of high noise levels. The paper mentions that the research area is typically quiet, with only other research vessels sometimes crossing the area. Would this then be representative of a reef system that has been exposed to noise, in which case these results can be seen as a positive impact of mitigation? Or do we actually see the baseline state in the quiet treatment and a diminished survival in the noise treatment? This distinction matters, because the effect of diminished noise levels on animals that have been frequently disturbed by noise in the past will likely be different than animals that have not been exposed to noisy environments previously. Lasting behavioural changes of repeated exposure have been shown before, for example by Nedelec et al. 2016, Harding et al. 2018, Kok et al. 2021. The authors need to give more information about baseline noise conditions in the area, as well as discuss chronic vs short-term noise exposure to fully back their claim.

Smaller comments:

L6 (abstract): add “fish” so that the sentence reads “survival of more fish offspring”

Figure 2: it took me several rereads to find out that “exposure” actually consisted of two types of exposure, depending on the long-term treatment the animals received. The method of comparing boat noise to ambient is sound, but it is confusing that exposure sometimes means exposure, and sometimes means sham-exposure. Is there a way to make this clearer in the figure?

L95-110: The authors show that hatchling length, but not weight, is larger in low-noise conditions than in high-noise conditions. This suggests that hatchlings in low-noise conditions are leaner. Why would this be the case? And could it be that not just length, but body/weight ratio influences survival rates?

Reviewer #3 (Remarks to the Author):

The current MS describes complementary field and lab experiments on the reproduction of a reef fish (spiny chromis). The results show that limiting boat traffic in the proximity of reefs increase fish larvae survival and also give insight into the mechanisms behind this. The experiments are clearly very well designed, the results are very interesting and complete, and the manuscript is very well written. A minor issue is that – for the field study – only one boat was used, which leads to pseudo replication (for the lab study this is not clear yet, I assume it was much easier to account for this here). But the experiments are so complete, logistically challenging and well designed, that this should not withheld publication in a good journal. I have very little comments, the most important is that some details of the lab study are lacking with limit the reproducibility of the study (all issues that caught my attention are in the comments below). I think this is all quite easy to add.

Specific comments:

L24-25: I'm not sure whether I agree with the first part of this sentence 'Our results demonstrate that noise mitigation could have population-level consequences'. As far as I'm aware, offspring survival has little impact on the population level. If the authors want to keep this statement in the summary, perhaps they can argue for it in the discussion.

L95-102: Perhaps the interaction with age can be explained by an exponentially faster growth with age.

L198-199: I think it would be nicer to refer to other studies that show effects of boat impact (instead of: 'There is no reason to suspect that limiting traffic noise would only benefit our study species')

L196-208: Personally, I feel that this paragraph is a bit out of place, even though I agree with the message. I do like figure 3 (perhaps with indication of what the height of the line means), but other than that the paragraph is very broad and has limited connection with the study. I would either make to connection/reference to the study much clearer and have more emphasis on reefs, or drop the paragraph. But of course, I also don't want to take ownership of the paper. So, if the editor (and maybe other reviewer(s)) do not make comments about it, it's also fine for me to keep the paragraph in.

L296: perhaps 'after hatching' instead of 'of hatching'?

L311-312: why was the 'settling time' variable? And 30 s of scoring sounds very short, can you argue that this is enough?

L345: Figure number is incorrect

L353: Perhaps remove 'back'

L373: Perhaps add some info like tank sizes and feeding regime to make the study more reproducible.

L409: Camera A & B are not really referring to a figure or so. I would rather label them a bit more informative, e.g.: top and side camera or so.

L419: Something is missing here.

L437: specify method for reproducibility.

L444: Perhaps structure like: T-tests were used where data was appropriate(?).

L467: Some info is lacking IMO. E.g. what kind of speakers, what playbacks (was accounted for pseudo replication), playback regime, perhaps rms SPL (and PM) values.

RESPONSE TO REVIEWERS' COMMENTS

We thank the reviewers for their constructive comments. We have addressed each of these and believe our manuscript is greatly improved as a result. Our responses are in *italics* and text that has been incorporated into the main text is **highlighted**.

Reviewer #1 (Remarks to the Author):

Noise has in recent years been shown to influence the fitness of species also in aquatic ecosystems, with an increasing number of studies showing effects especially on fishes. The aim of the present study was to determine if limiting boat traffic can mitigate the negative effects that the noise is known to have. The study design is thus the opposite to that usually applied when noise is added, and the results are – unsurprisingly - in line with earlier findings. The study consequently backs up earlier results on negative effects of noise on coral reef fishes and the importance of limited/regulating noise levels.

The study also attempts to determine the mechanism behind the effects of noise, through a more controlled laboratory experiment, but here the results are less clear and some overstatements are made (too strong conclusions are drawn regarding offspring survival and size).

We addressed this in our response to point 2) made by Reviewer #1.

More research should be done under these controlled conditions before any clear conclusions can be drawn and compared to the results from the field.

Again, we draw the editor's attention to our response to point 2) made by Reviewer #1. Given that we have now altered our wording to be careful to avoid overclaiming from results from the controlled conditions, we think it remains valuable to include these data as they point in the same direction as our data from the field and there is no indication that they contradict our conclusions drawn based on the field data. We agree further work in controlled conditions would be a welcome addition to the research area with more time and funding, as is always the case, but the benefits would need to be balanced with welfare considerations of removing more fish from the wild and keeping them in captivity. Our work in the laboratory reveals clear effects on length without the potential confounding effects of competition or predation (as are found in natural conditions) and thus we believe that this is a valuable component of this paper.

The main limiting factors of the present study, which need to be better taken into consideration in the paper are:

1) No information on noise levels at the protected 'limited boating reefs' in the absence of the protection, which would ensure that the 3 protected and the 3 control sites had the same noise levels before the experimental treatments started.

We do not anticipate any bias due to allocation of sites and have added the following to the methods (line numbers 267-270) in order to clarify:

"There is a navigable channel through the lagoon where the experiment was conducted. We chose six sites along the navigable route and randomly allocated these to treatment. Following random allocation, two sites were switched solely for safety reasons for motorboat drivers (Fig. S2)."

2) Some sample sizes are quite small and the results are not always clear cut (see for example line 88). Some of the conclusions drawn are consequently too strong.

On the basis of the three Rs we removed 30 pairs of spiny chromis from the wild. Of the 15 pairs randomly allocated to each treatment, 13 and 9 produced eggs. We present a simple analysis of this outcome carefully described as a trend, noting that this is congruent with the significant difference found in the larger number of pairs (84) studied in the field. The word similar has been changed to “congruent” in the main text (line number 87) to convey that the finding from the laboratory and the field are both pointing in the same direction.

3) It is not possible to separate between the effects of treatment and earlier experience (in the field) regarding the variables recorded in the laboratory experiment.

Adult spiny chromis pairs were randomly allocated to treatments so it is unlikely that there was a bias in prior experience in one treatment over another. However, if this were the case this would probably only weaken our results. The fish were initially all from one small area of reef at Lizard Island where all fish would be expected to have the same level of prior experience of motorboat noise. We have added the following to the methods to clarify (line numbers 369-376):

“Spiny chromis adults were caught with fine monofilament barrier nets and hand nets from the section of reef on the lagoon side of Palfrey Island within the lagoon around Lizard Island in the northern Great Barrier Reef (14° 41' S, 145° 27' E) during November 2016. All adults would have experienced equivalent prior noise exposure. The mean±SE standard length of adults was 10.7±0.1 cm. Spiny chromis were randomly allocated to treatments and were housed in 30 male–female pairs, and maintained at a mean±SE temperature of 27.7±0.1°C in the presence of either a busy-boating treatment (playback of motorboats in a pattern matching exposures in the field) or a no-boating treatment (playback of ambient reef sound).”

4) Only one species was investigated, which limits the generality of the findings.

We used a well-developed, fully tractable study system for the lab and field. In this system it is possible to study multiple life stages in the field and lab (adults and juveniles in the field and adults, embryos, juveniles in the lab). We embarked on a four-month field experiment encompassing the entire breeding season. Looking at more than one species at this level of detail and duration of study that we did was impractical for one investigation. Further work with other species in separate papers would add to an overall body of research but does not alter the fundamental primary findings of our investigation that reproductive output can be improved by limiting noise exposure – this has not been shown before.

Specific comments

Line 59-62. The lack of an effect of the treatment on the number of pairs producing offspring could be because of the short duration of the experiment before this variable was recorded (compared to the duration of the experiment before offspring survival was determined).

We thank the reviewer for this useful comment. If these treatments went on for longer we would expect to see even stronger effects. If an opportunity arises to look at longer term changes in exposure due to mitigation then we will follow up with this.

Line 84-85. How can survival be higher on the limited-boating reefs? More explanation needs to be given. On line 76 you indicate that more offspring were hatching on busy-boating than on limited-boating reefs, and on lines 81-81 that survival increased with hatch count on busy-boating reefs, but decreased on limited-boating reefs. This would result in many more offspring surviving on busy-boating reefs, as clutch size and number of breeding pairs did not differ – have I misunderstood something?

As field sites were checked every other day for new clutches, it is possible that some clutches suffered total mortality before they were observed. We have edited lines 79-81 and added a discussion point to lines 81-82:

“on limited boating reefs, survival was better for smaller broods than larger broods, whilst on busy-boating reefs smaller broods had lower survival than larger broods (Cox model: $X^2_1=40.79$, $p<0.001$). There was better offspring survival overall on limited-boating compared with busy-boating reefs ($X^2_1=34.87$, $p<0.001$; Table S1C; Fig. 1b). The smallest wild broods that hatched in the busy boating treatment may have suffered complete mortality before they were first observed due to their greater vulnerability in noise (5). This could explain the bias towards higher number of hatchlings counted at the first observation in the busy-boating treatment in the wild, not observed in the lab where counts of hatchings were guaranteed to occur within hours of hatching.”

Line 91-93. There was only a trend towards better survival in the lab experiment, and the sample sizes are very small so you cannot state that survival was higher. Be more careful in drawing your conclusions.

We have referred to this as a trend and altered the wording on the stated line (line number 92) to be careful not to overclaim on the basis of this result. We have deleted ‘and the no-boating treatment’.

Line 104. If treatment influenced offspring length but not weight it appears that the shape of the juveniles were different – do you have any more information on this? If weight was not influenced by treatment, it is also not true to state that the offspring were larger, they were only longer.

We thank the reviewer for this point, we have changed language used from larger to longer throughout. Length is the factor that is the most likely to impact capture by gape-limited predators, so our conclusions remain valid.

Line 360- If the individual were immediately exposed to the treatments in the lab, it is likely that the eggs clutch sizes had been determined before the fish were caught and exposed to the treatments. To be able to determine effects on clutch size and egg quality, you should have bred the fish in the lab under standardised conditions before you exposed them to the treatments. Given the present design, it is not possible to separate between the effects of the treatment and earlier experience in the field.

We thank the reviewer for the point that clutch size may have been determined prior to the acoustic conditions we exposed the fish to in the laboratory or the field. This may explain why we did not see responses in that measure. We randomly allocated sites in the field to treatments and pairs of fish to treatments in the laboratory to avoid any bias in prior experience. Conducting a multi-generation experiment would be interesting should the opportunity arise to look at multi-generational effects of

noise mitigation.

Reviewer #2 (Remarks to the Author):

The manuscript “Limiting motorboat noise on coral reefs boosts fish reproductive success” shows that spiny chromis fish have increased hatchling survival, as well increased hatchling size, in reefs with limited motorboat noise compared to reefs with frequent boat noise. The manuscript cleverly reframes the typical story line of how noise disturbance affects animals to one where decreased noise exposure (the old baseline) leads to improved fitness. The results of the manuscript are sound and the combination of studies in the field and in captivity enables the authors to identify the mechanisms that underlie the change in offspring survival between treatments.

We thank Reviewer #2 for this assessment.

Having said that, I do have some comments.

My main concern is the argumentation around improved fitness with noise mitigation. As mentioned before, the manuscript is framed in such a way that the focus is on improved hatchling survival in decreased noise conditions, rather than decreased survival in increased noise conditions.

During our study we are confident that we mitigated traffic noise at the limited boating reefs by marking them as areas to avoid on the research station map and monitoring the lagoon daily. We have added to the main text (line numbers 272-274):

“We limited the main source of traffic at the limited boating reefs by marking these reefs on the research station map as areas to avoid by at least 100 m and monitoring activity in the lagoon daily.”

I like the framing as a change in perspective for the field, but I miss some discussion on repeated or chronic noise exposure.

We limited the traffic and thus noise for the entire three-month breeding season. We have not used the specific terms ‘chronic’ or ‘short-term’ in our manuscript as these are subjective and would require definitions. We have instead provided accurate descriptions of the nature and duration of experimental conditions throughout the methods.

Showing effects of noise mitigation requires that the original state was one of high noise levels. The paper mentions that the research area is typically quiet, with only other research vessels sometimes crossing the area. Would this then be representative of a reef system that has been exposed to noise, in which case these results can be seen as a positive impact of mitigation?

We do not have data for the levels of traffic prior to the study but are aware that fishing boats, tourist boats and research station boats pass through the channel in the lagoon. The main source of traffic is research station boats. We have added the following to the methods in order to clarify (line numbers 268-272):

“There is a navigable channel through the lagoon where the experiment was conducted. Fishing boats, tourist boats and research station boats pass through the channel, but the main source of traffic is research station boats.”

Or do we actually see the baseline state in the quiet treatment and a diminished survival in the noise treatment?

As stated above, we are confident that we mitigated traffic noise at the limited boating reefs by marking them as areas to avoid on the research station map and monitoring the lagoon daily.

This distinction matters, because the effect of diminished noise levels on animals that have been frequently disturbed by noise in the past will likely be different than animals that have not been exposed to noisy environments previously.

This is a valid point that supports the conclusions we draw from our experimental manipulation.

Lasting behavioural changes of repeated exposure have been shown before, for example by Nedelec et al. 2016, Harding et al. 2018, Kok et al. 2021.

This is true. The examples cited would suggest that fish can increase their tolerance to noise with repeated exposure. Any effect of increased tolerance with prior exposure would reduce differences between our treatments, making our results most likely conservative. Previous work on the focal species, spiny chromis (Nedelec et al. 2016), as well as behavioural measures in the current study did not detect any increase in tolerance over 12 days of playback exposure, thus we did not discuss the likelihood that changes in tolerance might have reduced any differences in our manuscript.

The authors need to give more information about baseline noise conditions in the area

As detailed above, we have added the following additional information about baseline noise conditions in the area:

“There is a navigable channel through the lagoon where the experiment was conducted. Fishing boats, tourist boats and research station boats pass through the channel, with the main source of traffic being the research station boats.”

as well as discuss chronic vs short-term noise exposure to fully back their claim.

Smaller comments:

L6 (abstract): add “fish” so that the sentence reads “survival of more fish offspring”

We have made this change.

Figure 2: it took me several rereads to find out that “exposure” actually consisted of two types of exposure, depending on the long-term treatment the animals received. The method of comparing boat noise to ambient is sound, but it is confusing that exposure sometimes means exposure, and sometimes means sham-exposure. Is there a way to make this clearer in the figure?

We have clarified in the figure legend:

“Parental care in quiet conditions for each rearing condition compared with exposure to motorboat playback if reared in busy-boating (red) or a different ambient track if reared in no-boating (blue):”

L95-110: The authors show that hatchling length, but not weight, is larger in low-noise conditions than in high-noise conditions. This suggests that hatchlings in low-noise conditions are leaner. Why would this be the case? And could it be that not just length, but body/weight ratio influences survival rates?

It is usually considered the case that fish with a higher body weight to length ratio are in better condition (e.g. condition factors commonly used in the literature). However, we found that fish in the limited or no boating treatment were longer and also fish in the limited boating treatment survived

better, while we could not detect an effect on weight. It could be that selection is acting more strongly on length than on weight or condition factor in this species at this life stage. Weight is expected to be more variable as a metric depending on time since last feeding, amount eaten (which may be affected by sibling-sibling or sibling-parent competition) and rate of digestion and metabolism. A common technique to avoid effects of these confounding factors is to starve fish for 24 h prior to sampling, however with repeated sampling from the same clutch we were concerned that this could have affected growth in the rest of the brood. We thus have a clearer signal in length than weight. We have added the following to the discussion (line numbers 105-106):

“Weight as a measure of development may be subject to high variability due to food in the gut.”

Reviewer #3 (Remarks to the Author):

The current MS describes complementary field and lab experiments on the reproduction of a reef fish (spiny chromis). The results show that limiting boat traffic in the proximity of reefs increase fish larvae survival and also give insight into the mechanisms behind this. The experiments are clearly very well designed, the results are very interesting and complete, and the manuscript is very well written.

We thank Reviewer #3 for these comments.

A minor issue is that – for the field study – only one boat was used, which leads to pseudo replication (for the lab study this is not clear yet, I assume it was much easier to account for this here). But the experiments are so complete, logistically challenging and well designed, that this should not withheld publication in a good journal. I have very little comments, the most important is that some details of the lab study are lacking with limit the reproducibility of the study (all issues that caught my attention are in the comments below). I think this is all quite easy to add.

We thank the reviewer for these comments. There were eight boats and eight different drivers used in the field study and five different boats recorded for the lab study. This is detailed in the supplementary information. To improve readability we have altered the main text as follows (line numbers 276 and 375):

*“At these sites, we drove **eight different** 5 m aluminium motorboats”.*

*“Playback of **four of the five recorded** motorboats in a pattern matching exposures in the field.”*

Specific comments:

L24-25: I'm not sure whether I agree with the first part of this sentence 'Our results demonstrate that noise mitigation could have population-level consequences'. As far as I'm aware, offspring survival has little impact on the population level. If the authors want to keep this statement in the summary, perhaps they can argue for it in the discussion.

We respectfully disagree with the reviewer here. There is a rationale for offspring survival having population level consequences: Selection is acting very strongly at this life stage due to high mortality. Altering selective pressures will alter the phenotypes and genotypes that are able survive to adulthood. The genotypes that survive to adulthood in present day conditions are the best adapted to the multitude of other pressures, so altering this will mean that fewer individuals are able to survive until they are able to breed. Also, as the young fish are such an important food source, increasing the proportion that are eaten in one generation could lead to an imbalance between

predator and prey populations in following years – if predators can catch juveniles more easily and thus there is an increase in predator population size because of greater food availability, then unless the juvenile supply keeps growing (which is unlikely given the probable reduced numbers of individuals available for breeding) then predator populations may crash. The increased fluctuations in population sizes in the system could have unknown effects on the rest of the reef community due to changes in nutrient acquisition for the reef or other effects.

In light of this disagreement, we have softened the language in this phrase in the main text to the following (line numbers 24-25):

“Our results suggest noise mitigation could have benefits that carry through to the population level”

L95-102: Perhaps the interaction with age can be explained by an exponentially faster growth with age.

We agree with and thank the reviewer for this point, we have added the following to the text (line numbers 99-100):

*“We measured the standard length of up to 10 individuals per brood approximately weekly and found that limited-boating juveniles were longer than those from the busy-boating treatment; this effect became more apparent with age (LMM, treatment*age interaction: $X^2_1=8.97$, $p=0.003$; Table S1D; Fig. 1c), potentially due to exponentially faster growth with age.”*

L198-199: I think it would be nicer to refer to other studies that show effects of boat impact (instead of: ‘There is no reason to suspect that limiting traffic noise would only benefit our study species’)

We have referenced a recent review of the impacts of a wide variety of sources of anthropogenic noise on a wide variety of taxa (Duarte et al. 2021; Science).

L196-208: Personally, I feel that this paragraph is a bit out of place, even though I agree with the message. I do like figure 3 (perhaps with indication of what the height of the line means), but other than that the paragraph is very broad and has limited connection with the study. I would either make to connection/reference to the study much clearer and have more emphasis on reefs, or drop the paragraph. But of course, I also don’t want to take ownership of the paper. So, if the editor (and maybe other reviewer(s)) do not make comments about it, it’s also fine for me to keep the paragraph in.

We have added that building resilience is represented by shifting up the dashed line in the figure legend. We have also added a sentence (with a relevant reference) to contextualise our final paragraph (line numbers 198-199):

“Three pillars of action are required to rebuild coral reefs worldwide: 1) reducing climate threats, 2) reducing local threats and 3) restoration.”

L296: perhaps ‘after hatching’ instead of ‘of hatching’?

We thank the reviewer for this point and have made this change (line number 86).

L311-312: why was the 'settling time' variable? And 30 s of scoring sounds very short, can you argue that this is enough?

Settling time varied by a small amount relative to the total time. Variability was due to time taken to swim from the nest to the boat after camera placement at the nest. The amount of time for the survey is relatively equivalent to that which would be used for a predator survey by a snorkeller swimming 25 m (a standard survey method). The benefit of our videoing method is that we were able to use an observer that was blind to the experimental treatment and to focus on the nests as replicates.

L345: Figure number is incorrect

We have corrected this to 1b.

L353: Perhaps remove 'back'

We have removed this (line number 360).

L373: Perhaps add some info like tank sizes and feeding regime to make the study more reproducible.

This information is provided in the supplementary information (section 'Tank set up and conditions').

L409: Camera A & B are not really referring to a figure or so. I would rather label them a bit more informative, e.g.: top and side camera or so.

We have renamed these 'top camera' and 'side camera' throughout.

L419: Something is missing here.

We have removed the mistyped letters 'was u' (line number 420).

L437: specify method for reproducibility.

We have added 'from photos using ImageJ' to specify the method used to measure standard length (line number 444).

L444: Perhaps structure like: T-tests were used where data was appropriate(?).

We have edited this short section for clarity as follows (line numbers 453-456):

"For measures of parental care, or at the level of clutch, T-tests or Wilcoxon signed ranks tests were used. Where we measured several individuals from within multiple sites, clutches, or broods, linear mixed effects models (LMM) or generalized linear mixed effects models (GLMM) were used to control for the random effects of site, clutch, or brood, provided models fit the data satisfactorily."

L467: Some info is lacking IMO. E.g. what kind of speakers, what playbacks (was accounted for pseudo replication), playback regime, perhaps rms SPL (and PM) values.

This information is all provided in the supplementary information file under the subtitle 'Noise exposure regime'.

REVIEWERS' COMMENTS

Reviewer #2 (Remarks to the Author):

The improvements made to the manuscript and the authors' responses have adequately answered my questions and concerns. I have two points left.

1. I still think it be worthwhile to devote some text discussing how this study compares to other studies that have looked at chronic noise effects and noise mitigation. I agree with the authors that the definition of chronic noise or repeated noise exposure is arbitrary, but we can be certain that these animals were not naïve animals that were exposed to a short period of noise, as is the case for many other studies. Instead, they are part of a growing body of literature studying longer-term effects of noise. Furthermore, not all studies on noise mitigation find similar results to this study. Phillips et al 2021, for example, studied changes in noise exposure on seedling recruitment, and found that areas in which noise exposure was present initially but then stopped, did not recover to baseline levels. This is opposite to the findings of this study, albeit of course in a terrestrial system and on a different time scale.

2. The authors mention in their rebuttal that previous experiments (Nedelec et al 2016) have shown that their study species does not become more tolerant to noise with time. I would like to see this information added in the manuscript.

Reviewer #3 (Remarks to the Author):

I thank the authors for their clear response to my comments and the changes to their MS.

While examining the reply to my comments, I realised that I didn't provide my comment to L24-25. This is the paper I should have cited, which shows (based on modelling) that a high disturbance-induced reproductive failure is not very likely to have population consequences (doi: <http://dx.doi.org/10.1098/rspb.2020.0490>). This is - of course - a single modelling paper on another species, and the real world is likely more complex. But at least it shows to be careful in making bold statements (without proof/references) on whether juvenile mortality has population consequences.

Either way, I think the authors replied well to this specific comment, but I still wanted to share this.

There were two other things that caught my attention when I went through the MS again:"

L 74-75, In my opinion it's too bold to directly assume validity from lab studies to the field experiment.

L 129-170, perhaps it can be made clearer that these are the captive results

I did not see the supplement figures this time.

Either way, no response (to me) is required for this and I think the authors conducted a very nice and interesting study.

Reviewer #4 (Remarks to the Author):

Review of Nature Communications NCOMMS-21-26780A

I have been asked to comment specifically as to whether the concerns of reviewer #1 have been adequately addressed in the authors' response and also comment more generally on the methods used so have largely confined my review to those aspects, although I did read the entire revised manuscript and other reviewers' comments.

Reviewer #1 had 4 major concerns, as outlined below

Concern 1: There was no background sound levels taken at experimental sites before the study began.

This is a valid concern and would have been ideal but I do not feel it is a fatal flaw in the current manuscript. I have been to the study site, albeit some years ago, and there is a fairly small channel through which boats traverse so random allocation of these sites as described herein would be expected to encompass sites with similar levels of background boat traffic before the study commenced. Future studies, especially those over wider areas, should certainly have the background levels properly assessed before the study commences but given the somewhat unique nature of this small channel I do not feel there would be significant differences in noise signatures.

Concern 2: sample sizes, especially in the lab setting, are too small

Again a valid concern but given the authors now rewording the possibility of small sample sizes masking effects have been addressed. There is debate in this field as to the validity of any lab experiments, due to acoustics issues in small tanks, but by comparing field and laboratory effects in the current paper the authors use the lab experiments to isolate noise vs other effects as explanatory variables. The results between field and lab are largely corroborated in the current study and the authors are now careful to not overstate their lab results. Also, the study took place in a globally protected marine reserve so they were constrained on how many fish they could remove from the wild for their lab experiments

Concern 3: It was not possible to separate the effects of early exposure in the field before being brought into the lab.

Again because of the narrow channel from which fish were collected and the random allocation of fish to treatment it is unlikely there was a bias inherent in the laboratory fish.

Concern 4: Only one species was investigated.

This concern remains an issue in this field in general and some investigators have taken to using a more natural community approach but there is still value in the single species approach used here. To date the largest response differences between groups of fish on noise exposure trial is between groups with different degrees of hearing specialization and there is little evidence of species-specific responses with a given group. While expanding our species pool is always desirable, and Lizard Island Chromis species are perhaps overrepresented, the other side of that same coin is that a study with multiple species might not be able to have the same level of life history coverage as the current experiment.

My overall impressions of the manuscript and the methods used is quite favourable. As the reviewers make note of, this is one of the first papers to examine the possible effects of noise mitigation on fish growth and survival and the care taken in the field experiments is to be commended. I would have preferred a more careful analysis of sound exposures before the trials started but, again, there were likely few differences in boat traffic over sites in such a small area. I also appreciate their use of laboratory trials to complement the field approach, as both approaches have value, and while large sample sizes are always encouraged it is important to balance that against the conservation concerns of the species and area under study. This is a valuable paper to the field and it was nice to see attempted solutions to the problem of increasing underwater noise levels rather than just another paper showing noise is harmful.

Response to reviewers' comments

Reviewer #2 (Remarks to the Author):

1. I still think it be worthwhile to devote some text discussing how this study compares to other studies that have looked at chronic noise effects and noise mitigation. I agree with the authors that the definition of chronic noise or repeated noise exposure is arbitrary, but we can be certain that these animals were not naïve animals that were exposed to a short period of noise, as is the case for many other studies. Instead, they are part of a growing body of literature studying longer-term effects of noise. Furthermore, not all studies on noise mitigation find similar results to this study. Phillips et al 2021, for example, studied changes in noise exposure on seedling recruitment, and found that areas in which noise exposure was present initially but then stopped, did not recover to baseline levels. This is opposite to the findings of this study, albeit of course in a terrestrial system and on a different time scale.

We have added material on other studies that have looked at chronic noise effects and noise mitigation to the penultimate paragraph (lines 177–188):

“There is no reason to suspect that limiting traffic noise would only benefit our study species: **chronic and acute** anthropogenic noise compromises behaviour, physiology, reproduction and survival in a range of **marine and terrestrial** organisms, while natural, unpolluted soundscapes are key to settlement, recruitment **and other ecological functions (3)**. It is true that some fishes **show increased tolerance to noise disturbance when faced with repeated exposure (e.g., 25-27), but this is not seen in all, including our study species (e.g., 13, 28)**. There is mixed evidence from terrestrial systems on ecological recovery in quieter conditions (e.g., **birdsong in lockdown increased (28), while seed dispersal around disused gas wells did not (30)**) but, in **general, reproductive** success and survival are strongly linked to population stability (31). **In fish in particular, improvements in energy intake and energy expenditure could be important drivers of population growth via size-dependent fecundity (32)**. Therefore, limiting traffic noise could help conserve and restore healthy populations of coral reef fishes.”

2. The authors mention in their rebuttal that previous experiments (Nedelec et al 2016) have shown that their study species does not become more tolerant to noise with time. I would like to see this information added in the manuscript.

We have incorporated this into our edits to the penultimate paragraph (lines 180-181):

“some fishes show increased tolerance to noise disturbance when faced with repeated exposure (e.g., 25-27), but this is not seen in all, including our study species (e.g., 13, 28).”

Reviewer #3 (Remarks to the Author):

While examining the reply to my comments, I realised that I didn't provide my comment to L24-25. This is the paper I should have cited, which shows (based on modelling) that a high disturbance-induced reproductive failure is not very likely to have population consequences (doi: <http://dx.doi.org/10.1098/rspb.2020.0490>). This is - of course - a single modelling paper on another species, and the real world is likely more complex. But at least it shows to be

careful in making bold statements (without proof/references) on whether juvenile mortality has population consequences.

We thank the reviewer for this modelling study. This model did not include any parameters to reflect the hypothesis that when small numbers of offspring survive to reproduce due to strong selective pressure, small shifts in selective pressures have the potential to further narrow the bottleneck of number of individuals that can contribute to the next generation and some advantageous adaptations could thus be lost. However, what this study does show is that food intake and energy expenditure could potentially have a strong impact on population-level consequences via impacts on growth due to size-dependent fecundity. Our findings that growth and activity were improved on limitation of noise exposure thus aligns well with our comment in the paper “Our results suggest noise mitigation could have benefits that carry through to the population level” and the reference provided, which we cite (line 184).

To support this point, we have added the following in green to our Abstract (lines 29–32):

“Our results suggest noise mitigation could have benefits that carry through to the population level by increasing adult reproductive output and offspring growth, thus helping to protect coral reefs from human impacts and presenting a valuable opportunity for enhancing ecosystem resilience.”

And added an emphasis on size (185–188):

“In fish in particular, improvements in energy intake and energy expenditure could be important drivers of population growth via size-dependent fecundity (32). Therefore, limiting traffic noise could help conserve and restore healthy populations of coral reef fishes.”

Either way, I think the authors replied well to this specific comment, but I still wanted to share this.

There were two other things that caught my attention when I went through the MS again:"
L 74-75, In my opinion it's too bold to directly assume validity from lab studies to the field experiment.

We state that the lab study is ‘complementary’ to the field study (lines 27, 79, 170). In none of the measures that we made did we find evidence in the lab that contradicted the evidence from the – more ecologically valid – field study. This supports the supposition that noise, rather than any other factor, could be driving the responses we saw in the wild. The lab study was also not constrained by limitations on number of sites in the field – so any concern that pseudoreplication could be driving the responses from the field are assuaged.

L 129-170, perhaps it can be made clearer that these are the captive results.

We state the following and have added the word ‘captive’ to emphasise this point (lines 81–84, 128–128, 147–150):

“Our laboratory-based experiment allowed isolation of noise as the causal factor of impacts on reproductive success and, since eggs are naturally laid in caves within the reef, assessment

of effects on clutch characteristics, egg development and parental-care investment that were unobservable in the wild.”

“To consider direct effects at the embryonic stage, we sampled 10 eggs per **captive** clutch within 2 h of laying.”

and

“On day 10 of embryonic development, we examined egg-fanning and activity (as a proxy for guarding) of **captive** parents before and during exposure to either motorboat-noise playback (in the busy-boating treatment) or a different ambient-reef playback (in the no-boating treatment).”

I did not see the supplement figures this time.

Now included.

Either way, no response (to me) is required for this and I think the authors conducted a very nice and interesting study.

Reviewer #4 (Remarks to the Author):

Review of Nature Communications NCOMMS-21-26780A

I have been asked to comment specifically as to whether the concerns of reviewer #1 have been adequately addressed in the authors’ response and also comment more generally on the methods used so have largely confined my review to those aspects, although I did read the entire revised manuscript and other reviewers’ comments.

Reviewer #1 had 4 major concerns, as outlined below

Concern 1: There was no background sound levels taken at experimental sites before the study began.

This is a valid concern and would have been ideal but I do not feel it is a fatal flaw in the current manuscript. I have been to the study site, albeit some years ago, and there is a fairly small channel through which boats traverse so random allocation of these sites as described herein would be expected to encompass sites with similar levels of background boat traffic before the study commenced. Future studies, especially those over wider areas, should certainly have the background levels properly assessed before the study commences but given the somewhat unique nature of this small channel I do not feel there would be significant differences in noise signatures.

Concern 2: sample sizes, especially in the lab setting, are too small

Again a valid concern but given the authors now rewording the possibility of small sample sizes masking effects have been addressed. There is debate in this field as to the validity of any lab experiments, due to acoustics issues in small tanks, but by comparing field and

laboratory effects in the current paper the authors use the lab experiments to isolate noise vs other effects as explanatory variables. The results between field and lab are largely corroborated in the current study and the authors are now careful to not overstate their lab results. Also, the study took place in a globally protected marine reserve so they were constrained on how many fish they could remove from the wild for their lab experiments
Concern 3: It was not possible to separate the effects of early exposure in the field before being brought into the lab.

Again because of the narrow channel from which fish were collected and the random allocation of fish to treatment it is unlikely there was a bias inherent in the laboratory fish.

Concern 4: Only one species was investigated.

This concern remains an issue in this field in general and some investigators have taken to using a more natural community approach but there is still value in the single species approach used here. To date the largest response differences between groups of fish on noise exposure trial is between groups with different degrees of hearing specialization and there is little evidence of species-specific responses with a given group. While expanding our species pool is always desirable, and Lizard Island Chromis species are perhaps overrepresented, the other side of that same coin is that a study with multiple species might not be able to have the same level of life history coverage as the current experiment.

My overall impressions of the manuscript and the methods used is quite favourable. As the reviewers make note of, this is one of the first papers to examine the possible effects of noise mitigation on fish growth and survival and the care taken in the field experiments is to be commended. I would have preferred a more careful analysis of sound exposures before the trials started but, again, there were likely few differences in boat traffic over sites in such a small area. I also appreciate their use of laboratory trials to complement the field approach, as both approaches have value, and while large sample sizes are always encouraged it is important to balance that against the conservation concerns of the species and area under study. This is a valuable paper to the field and it was nice to see attempted solutions to the problem of increasing underwater noise levels rather than just another paper showing noise is harmful.

We thank reviewer #4 for these comments.